# A Journey through the Inter-Cellular Interactions in the Bone Marrow in Multiple Myeloma: Implications for the Next Generation of Treatments

**DOI:** 10.3390/cancers14153796

**Published:** 2022-08-04

**Authors:** Rosario Hervás-Salcedo, Beatriz Martín-Antonio

**Affiliations:** Department of Experimental Hematology, Instituto de Investigación Sanitaria-Fundación Jiménez Diaz (IIS-FJD), University Autonomous of Madrid (UAM), 28040 Madrid, Spain

**Keywords:** multiple myeloma, bone marrow, marrow-infiltrating lymphocytes, mesenchymal stromal cells, cell–cell interactions

## Abstract

**Simple Summary:**

Here, we describe the main interactions of multiple myeloma cells in the bone marrow with non-hematological and hematological immune cells that impact the disease’s progression and treatment resistance. Non-hematological cells in the bone marrow can secrete molecules that accelerate disease progression. On the other side, these interactions compromise immune cell activity, leading to immune evasion and disease progression. Deep knowledge of these interactions can lead to the design of improved treatments for multiple myeloma.

**Abstract:**

Tumors are composed of a plethora of extracellular matrix, tumor and non-tumor cells that form a tumor microenvironment (TME) that nurtures the tumor cells and creates a favorable environment where tumor cells grow and proliferate. In multiple myeloma (MM), the TME is the bone marrow (BM). Non-tumor cells can belong either to the non-hematological compartment that secretes soluble mediators to create a favorable environment for MM cells to grow, or to the immune cell compartment that perform an anti-MM activity in healthy conditions. Indeed, marrow-infiltrating lymphocytes (MILs) are associated with a good prognosis in MM patients and have served as the basis for developing different immunotherapy strategies. However, MM cells and other cells in the BM can polarize their phenotype and activity, creating an immunosuppressive environment where immune cells do not perform their cytotoxic activity properly, promoting tumor progression. Understanding cell–cell interactions in the BM and their impact on MM proliferation and the performance of tumor surveillance will help in designing efficient anti-MM therapies. Here, we take a journey through the BM, describing the interactions of MM cells with cells of the non-hematological and hematological compartment to highlight their impact on MM progression and the development of novel MM treatments.

## 1. Introduction

Nowadays, it is widely accepted that the tumor microenvironment (TME) is a relevant component in tumors that modulates the response to cancer treatments affecting tumor progression. The TME consists of an extracellular matrix, a plethora of tumor cells, and a variety of non-tumor cells with complex interactions. These interactions, either through cell–cell contact or as soluble mediators, can accelerate tumor progression and the lack of response to cancer therapy [1]. Moreover, the knowledge of these interactions enables the development of non-immunotherapy [2,3,4] and immunotherapy strategies [5,6,7,8,9] in cancer patients.

Non-tumor cells in the TME, including endothelial cells, fibroblasts, and immune cells [7], modulate the responses to chemotherapy cancer treatments. For instance, chemotherapy agents that induce DNA damage, such as doxorubicin, trigger cytokine production by endothelial cells that decrease chemosensitivity of tumor cells to these treatments [10]. DNA-damaging agents also induce a senescence state in cells with the production of a senescence-associated secretory phenotype (SASP), a secretome rich in chemokines and growth factors that promote tumor progression [11]. Indeed, the secretion of SASP by endothelial cells in the TME includes IL6 secretion and chemoresistance development [12]. Tumor-associated macrophages (TAMs) with an M2-like phenotype provide a survival advantage to tumor cells in hypoxic conditions through IL6 receptor-mediated signals [13]; they protect tumor cells against paclitaxel, etoposide, and doxorubicin [14]. Moreover, platinum-based therapy supports monocyte differentiation to M2 macrophages, which associates with tumor progression [15].

Cellular components in the TME also influence the efficacy of radiotherapy treatments. Hence, radiotherapy activates fibroblasts, which become cancer-associated fibroblasts (CAFs). While some studies argue that CAFs promote tumor progression, others claim they are beneficial [16,17]. Thus, CAFs can secrete cytokines, such as IL32 that promote cancer cell invasion and metastasis [18]. However, CAFs in vivo depletion accelerates pancreatic cancer accompanied by epithelial-to-mesenchymal transition and enhanced T-regulatory (regs) cells that is reversed with anti-CTLA4 immunotherapy [19].

Immune cells and their secretome also shape the TME [1], impacting cancer progression and the efficacy of immunotherapy treatments [20]. For instance, tumor-infiltrating cells (TILs) in the TME are the basis for developing immunotherapy strategies based on immune checkpoint inhibition (ICI) that try to reactivate the tumor immune-surveillance activity of TILs [9]. Radiotherapy can promote tumor-specific immunity by activating dendritic cells (DCs) in the TME that support tumor-specific effector CD8 T cells [21]. Moreover, immunotherapy strategies based on the infusion of chimeric antigen receptor (CAR)-modified T cells have significantly improved the treatment of hematological malignancies [22,23,24,25]. However, in solid tumors, the barriers imposed by the TME [26] have delayed the development of efficient CAR-T cell therapies. Age also seems to play an essential role in the immune cells’ activity and, therefore, in immunotherapy. Thus, in hematological malignancies, pediatric patients with acute lymphoblastic leukemia (ALL) have achieved outstanding responses after treatment with CAR-T cells [22]. However, in adult patients with multiple myeloma (MM) [27], a disease where aging is a risk factor and where the TME is more relevant than in ALL, a proportion of patients end-up relapsing. In MM, the progression of the disease is drastically affected by the TME, either by soluble factors or cell–cell interactions in the bone marrow (BM) [28]. Moreover, relapses after administration of CAR-T cells [27], and the lack of efficacy of ICI therapies with significant toxicities in MM [25] might be partly explained by the impact of non-immune and immune cell interactions in the TME.

Here, we will take a journey through the BM, describing the interactions of plasma MM cells with cells of the non-hematological compartment to highlight the impact that these interactions have on the survival of tumor cells and the development of novel MM treatments. Moreover, we will describe the main differences found in the different immune cell subsets in the BM of MM patients that might lead to deficient tumor surveillance and failure of immunotherapy treatments in MM.

## 2. Impact of Interactions between Non-Hematological Cells and MM Cells in the BM

### 2.1. Extracellular Matrix (ECM)

MM is a hematologic malignancy characterized by clonal proliferation of plasma cells in the BM [29]. However, different trafficking events of MM cells allow them to reach distinct niches from the BM, re-circulate to the BM, and finally egress from the BM during the extramedullary stage of the disease [28]. When MM cells re-enter the BM, they use the BM sinusoids, where the interaction CXCR4/CXCL12 is critical to promote both MM cell homing and retention in the BM [30]. In the BM, MM cells will interact first with proteins in the ECM, a complex layer of proteins that serves as a scaffold for many cells. Interactions between MM cells and the ECM are required for cell proliferation, migration, and survival [31]. Specifically, CD138 and VLA-4 on MM cells directly interact with the ECM proteins, such as collagen type 1 and fibronectin. The binding of VLA-4 to fibronectin induces activation of nuclear factor-kB (NFkB), inducing tumor cell survival and cell adhesion-mediated drug resistance [32]. These interactions generate a welcome and growth-supporting environment that stimulates the dissemination of the malignant plasma cells and results in the upregulation of anti-apoptotic proteins and cell cycle dysregulation [33]. Strategies used in the clinic to disrupt these MM–ECM interactions and reduce cell adhesion-mediated drug resistance include the CXCR4 inhibitor AMD3100 or the proteasome inhibitor bortezomib, which downregulates VLA-4 on MM cells [34], leading to the de-adhesion of MM cells from the BM and turning them more sensitive to therapeutic agents [35]. However, although these agents can enhance the efficacy of treatments by disrupting these interactions, they also contribute to the mobilization of MM cells from the BM into the circulation, promoting extramedullary disease [36].

### 2.2. Control of the Stroma by BM Mesenchymal Stromal Cells (BM-MSCs)

In physiological conditions, the primary cell population in the BM stroma, known as bone marrow mesenchymal stromal cells (BM-MSCs), support the maintenance and differentiation of hematopoietic lineages, regulate bone homeostasis and contribute to the spatial delimitation of the endosteal and vascular niches [37]. However, in MM, BM-MSCs, as part of the BM microenvironment, play a crucial role in the pathology of the disease. Despite being at low proportions in the BM (0.01 to 0.001% of mononuclear cells) [38], BM-MSCs are the main population among BM stromal cells that interact with MM cells by direct cell–cell contact or through paracrine secretion of different pro-survival cytokines. Thus, for instance, binding VLA-4 on MM cells to VCAM-1 on BM-MSCs promotes activation of NFkB increasing MM cell survival and proliferation [39]. Moreover, the integrin lymphocyte function-associated antigen 1 (LFA-1) on MM cells and its transmembrane binding partner Mucin 1 (MUC1) bind to ICAM-1 in adjacent BM-MSCs, resulting in the activation of different pathways associated with poor prognosis and disease progression in patients [40]. The strong impact of the interactions with BM-MSCs in the physiology of MM cells and their acquisition of multidrug resistance phenotype justifies their consideration as targets for MM therapy. Indeed, some drugs have been developed to disrupt these interactions and tested in MM patients, such as Natalizumab, a recombinant humanized IgG4 monoclonal antibody (MoAb) which binds α4 integrin impairing the interaction VLA-4/VCAM-1 (NCT00675428). Other promising approaches have been preclinically evaluated, such as the LFA-1 inhibitor LFA878 [41].

Soluble mediators are also required for MM plasma cell survival and proliferation in the BM. Thus, MM cells induce BM-MSCs to secrete cytokines that will be used for their benefit. Specifically, the main secreted cytokine is interleukin-6 (IL6), which is involved in MM growth, survival, migration, and drug resistance [42]. In turn, MM cells use IL6 to enhance the secretion of vascular endothelial growth factor (VEGF) and basic fibroblast growth factor (bFGF). Then, both VEGF and bFGF bind to their receptors on BM-MSCs, re-stimulating IL6 production [43]. Whereas inhibition of IL6 has not shown clinical benefit in MM [44], blocking of IL6 receptor with tocilizumab has shown efficacy in MM patients [45]. Furthermore, MM cell interactions with BM-MSCs cells are mediated through Notch pathways and Dickkopf-1 (DKK1), which induce the secretion of IL6, VEGF, and insulin-like growth factor (IGF-1) in BM-MSCs [46,47]. Moreover, MSC-derived exosomes contain the long intergenic noncoding RNA LINC00461, which promotes MM cell proliferation and suppresses the beneficial effect of dexamethasone treatment. Indeed, the knockdown of LINC00461 enhances the beneficial impact of dexamethasone in preclinical studies [48].

B-cell activating factor (BAFF) and a proliferation inducing ligand (APRIL) are additional mediators with a protective effect on MM cells [49]. BAFF is a member of the tumor necrosis factor (TNF) family expressed on the surface of BM-MSCs and as a soluble form. BAFF stimulates B cell growth, and ligation of BAFF leads to increased proliferation and survival of MM cells [50]. APRIL is a secreted protein by BM-MSC that binds to B-cell maturation antigen (BCMA) and to transmembrane activator and calcium-modulator and cyclophilin ligand (TACI) on MM cells [51]. Therefore, APRIL-based CARs target MM cells expressing either BCMA or TACI with high efficacy at pre-clinical levels [52]. Moreover, BM-MSCs also protect MM cells against the lytic machinery of CAR-T cells [53].

Another member of the TNF family involved in this stromal training is TNFα, which induces the expression of adhesion molecules, such as LFA-1, ICAM-1, VCAM-1, and VLA-4 on MM cells, as well as ICAM-1 on BM-MSCs, resulting in increased binding of MM cells to BM-MSCs and further enhancing IL6 secretion [54]. These paracrine loops are critical for maintaining the constant growth of MM cells through the activation of different signaling pathways. In addition, MM cell interactions with BM-MSCs, added to the senescent status of cells in the BM in MM, further enhance the secretion of cytokines, chemokines, and soluble factors secreted by BM-MSCs to the BM milieu, which induce further MM proliferation and survival. TNFα, crucial in inflammation, is related to bone resorption and enhanced in MM patients. Thus, targeting TNFα could improve MM responses to treatments [55]. However, reports in inflammatory diseases suggest that anti-TNF-α inhibitors enhance the risk of having future hematological malignancies [56].

In summary, these interleukins and growth factors secreted by BM-MSCs cause tumor growth and drug resistance, limiting current MM treatments’ impact. Indeed, they are promising targets for developing anti-MM therapies that avoid the negative effect of BM-MSCs on dexamethasone treatment [48], on CAR-T cell therapies [53], or the negative impact of IL6 secretion. Thus, tocilizumab, an anti-IL6R [57], BHQ880, a monoclonal antibody against DKK1 [58], or tabalumab, a potent and selective fully human IgG4 MoAb with neutralizing activity against membrane-bound and soluble BAFF [59] are strategies that could be added to MM treatment.

### 2.3. Osteoclast/Osteoblast Imbalance in the Endosteal Niche

As previously mentioned, MM cells not only interact with the stromal compartment they also alter the endosteal and vascular niches in the BM. In the endosteal niche, healthy bone remodeling in the BM is maintained by a balance between bone formation (osteoblastogenesis) versus bone degradation (osteoclastogenesis). However, MM cells alter this dynamic balance, enhancing bone resorption to enable space for MM proliferation, causing the osteolytic lesions characteristic of myeloma bone disease (MBD) in around 80–90% of MM patients [60]. The negative impact of MBD on patient survival, quality of life, and public health costs has led to the development of different approaches to block MM-endosteal niche interactions. Strategies for patients to treat and avoid MBD have recently been reviewed [61]. Here, we describe which interactions of MM cells with cells in the endosteal niche, including BM-MSCs and other bone populations, such as osteoclasts and osteoblasts, accelerate MBD.

MM cells, through different mechanisms, upregulate osteoclast activity and differentiation resulting in imbalanced bone resorption, causing the osteolytic lesions of the MBD [62]. Specifically, MM cells secrete macrophage inflammatory protein-1α (MIP1α) and MIP1β that directly activate osteoclast formation and activity [63,64]. In turn, osteoclasts secrete IL6 to stimulate their self-proliferation and the proliferation of MM cells [65]. This interaction upregulates Chondroitin synthase 1 (CHSY1), which induces Notch signaling promoting MM cell survival and stimulating the recruitment of osteoclast precursors to increase bone resorption [66]. Macrophage-colony stimulating factor (M-CSF) and receptor activator of NFkB (RANK) ligand (RANKL) are additional factors required for osteoclast differentiation. Osteocytes produce RANKL, which promotes osteoclast activity through binding to RANK on osteoclastic lineage cells [67]. Nevertheless, MM cells’ interaction with BM-MSCs leads to the secretion of RANKL by BM-MSCs, further stimulating osteoclast activation and differentiation and enhancing bone lysis. This interaction also leads to the production of cytokines by BM-MSCs, such as IL6, which further promotes osteoclast growth [68]. In this way, amino-bisphosphonates have been administered in MM patients as first-line therapy for MBD due to their capacity to inhibit osteoclast activity [69]. Moreover, Denosumab, a fully human monoclonal antibody against RANKL, has also shown clinical benefit in MM patients [70]. Denosumab inhibits the development and activity of osteoclasts, decreases bone resorption, and increases bone density [71].

On the other hand, MM cells prevent osteoblast progenitor cell maturation and inhibit osteoblast activation, to continue impairing bone formation. Direct cell–cell interactions of MM cells through binding to VCAM-1 on osteoblast progenitors downregulate RUNX2 activity, essential for osteoblast differentiation [72]. Moreover, osteoblasts and BM-MSCs produce osteoprotegerin (OPG), which prevents the development of bone alterations caused by osteoclast/osteoblast imbalance. However, the binding of VLA-4 on MM cells to VCAM-1 on BM-MSCs decreases OPG secretion, forcing the balance in favor of osteoclasts and bone degradation [73]. Disrupting this VLA-4/VCAM-1 interaction with monoclonal antibodies, such as Natalizumab, could prevent bone lysis in MM patients, as described in preclinical models [74]. On the other hand, BHQ880, the DKK1-neutralizing antibody, can increase osteoblast differentiation, blocking the negative effect of MM cells on osteoblastogenesis and reducing IL6 secretion in MM patients [75].

### 2.4. Angiogenesis Promotion in the Vascular Niche

During the development of MM, an alteration in the neovascularization process occurs that affects the vascular niche. Neovascularization is the formation of new vessels from existing ones through endothelial cells (angiogenesis) or from endothelial precursors (vasculogenesis). Interactions between plasma cells and the BM microenvironment can modify this biological process [76,77,78].

Angiogenesis in cancer involves an early balanced avascular phase that gives rise to an uncontrolled and unlimited in-time vascular phase [79]. In the context of MM, Rajkumar et al. demonstrated that the BM microvascular density is increased in MM patients [80]. In this environment, the accumulation of MM cells in the BM generates hypoxic tumors highly expressing hypoxia-inducible factor-1 alpha (HIF-1α). HIF-1α will upregulate angiogenesis to deliver oxygen and nutrients and remove catabolites [81]. Different cytokines control angiogenesis, such as VEGF, fibroblast growth factor-2 (FGF-2), and hepatocyte growth factor (HGF). In MM, MM plasma cells become CD45-negative and produce VEGF [82]. Moreover, endothelial cells in the BM of MM modify their phenotype, expressing surface receptors related to angiogenesis, such as VEGFR-2 and Tie2/Tek, and increased expression of the β3-integrin and endoglin [83]. This differentiated phenotype in endothelial cells of the BM enhances MM cell interaction with the new-formed blood vessels and favors the entry and dissemination of MM cells into the circulation. This angiogenic phenotype in MM cells can also be induced by oncogenes, such as C-MYC, C-FOS, C-JUN, and ETS-1, which become active as a consequence of the genetic instability and immunoglobulin translocations in MM [84].

On the other hand, the differentiation of endothelial progenitors termed angioblasts during embryogenesis causes the development of the vascular system, known as vasculogenesis [85]. Studies suggest that vasculogenesis is responsible for the neovascularization in the BM in MM [86,87]. Indeed, endothelial markers such as VIII-related antigen (FVIII-RA), vascular endothelial-cadherin (VE-cadherin), VEGFR-2, TIE/Tek, and CD133 are expressed in endothelial cells of the neovessel wall [88]. Moreover, interactions of MM cells with BM-MSCs in the BM also impact vasculogenesis. Thus, MM cells stimulate BM-MSCs in the vascular niche to secrete HGF, VEGF, and IL8, further inducing neovascularization [89]. In turn, endothelial cells in MM will produce IGF1 and IL6 to promote MM cell growth, causing an autocrine loop in endothelial cells, which will enhance their production of VEGF, platelet-derived growth factor (PDGF), Ang-1, HGF, and IL1 to promote angiogenesis constantly [90].

The relevance of angiogenesis in the development of MM has led to the development of different treatments targeting this process. For instance, amino-bisphosphonates that inhibit osteoclasts also present anti-angiogenic activities and are administered in MM patients as supportive therapy for bone disease [69]. Ria et al. reviewed different strategies in MM mainly based on VEGF inhibition, such as monoclonal antibodies anti-VEGF (Bevacizumab) [91]. However, the addition of bevacizumab to anti-MM therapies did not result in a significant improvement in the outcome of patients [92,93]. Derivatives of quinolone and quinazoline, which inhibit a variety of tyrosine kinases, including VEGFRs, EGFR, and PDGFR have also been tested in MM patients. Despite their in vitro activity and reduced plasma levels of VEGF in treated MM patients, no responses or clinical benefits were achieved [94,95]. These disappointing results inhibiting a single proangiogenic cytokine could be related to the role played by hypoxia and other active pro-angiogenic pathways in the BM microenvironment, and greater efficacy could be feasible with drugs that simultaneously block multiple cytokines. Moreover, immunomodulators (IMIDs), such as thalidomide or lenalidomide, have revealed anti-angiogenic activity and inhibition of the secretion of angiogenic cytokines in MM patients [96,97].

To summarize, interactions between MM cells and cells that do not belong to the hematological BM compartment form a feedback loop that leads to bone destruction, angiogenesis, and tumor expansion in the BM. Unraveling these interactions has allowed the development of novel treatments for MM patients combining different strategies to block several molecules simultaneously. The most pertinent inter-cellular interactions, their impact, and the possible options of treatments are summarized in Figure 1 and Table 1. However, as we will explain in the next section, immune cells also play a relevant role in MM progression.

## 3. Impact of Interactions between Immune Cells and MM Cells in the BM of MM Patients

On the other hand, immune cells make up the hematological compartment of the BM, where the crosstalk between MM plasma cells and immune cells, from both myeloid and lymphoid lineage, plays a critical role in MM growth and maintenance. However, while non-hematological cell populations are partners that MM cells use for their benefit, the immune cell compartment is naturally composed of anti-tumor cells involved in tumor immunosurveillance [149]. However, MM cells create an immunosuppressive microenvironment and evade the immune system, a dynamic process encompassing multiple aspects of tumor cell–immune cell interactions [150] that favors MM proliferation and resistance to treatments [151]. In addition, the immune cells’ plasticity and capacity to polarize to different subsets enable the survival of normal and malignant plasma cells in the BM [152]. Another level of complexity comes with the deterioration of the immune system associated with aging, where MM represents an elderly population. Moreover, the accumulation and/or recruitment of immunosuppressive lymphoid and myeloid cells within the BM is another powerful mechanism during myelomagenesis [153].

This section describes relevant aspects of the different immune cell subsets in MM patients related to their inability to control MM progression. These interactions and how to avoid their detrimental impact are summarized in Figure 2 and Table 1.

### 3.1. Effector CD8 T Lymphocytes

The potential anti-MM activity of T cells in MM was suggested in the past with the observation that levels of CD3, CD4, CD8, and CD19 cell subsets in MM are associated with response to chemotherapy and survival [154]. Thus, the presence of T cell clones in MM associates with prolonged overall survival suggesting their anti-tumor activity [155]. Indeed, tumor-infiltrating lymphocytes (TILs), defined as the lymphocytic cell populations present in tumors, can be used to design adoptive cellular immunotherapy strategies [156]. In MM, TILs are defined as marrow-infiltrating lymphocytes (MILs). Adoptive transfer of MILs in MM patients demonstrated a direct correlation between tumor specificity of the MILs with clinical outcomes. Of interest, compared to peripheral blood (PB) lymphocytes, MILs show increased expression of CXCR4, which through CXCL12/CXCR4 could facilitate the trafficking of MILs to the BM [101]. An interesting approach to enhance the trafficking of MILs could be to modify them to over-express CXCR4 [104]. In MM patients receiving MILs, CD8 T cells are the main cytotoxic T cell subset. It was observed that a central memory (CM) phenotype in CD8 T cells of MILs at baseline was associated with achieving complete responses (CR), whereas patients with disease progression had a higher frequency of terminally differentiated effector T cells at baseline [101]. An approach to avoid terminally differentiated MILs could be adding a PI3K inhibitor during the production of MILs, that in CAR-T cells has shown efficacy in reducing the proportion of highly differentiated or senescent T cells [105].

Moreover, CD8 T cells can become exhausted due to prolonged antigen stimulation [157]. Exhausted T cells express immune checkpoint receptors, including programmed death-1 (PD-1), cytotoxic T-lymphocyte-associated antigen 4 (CTLA4), LAG-3, or T cell immunoglobulin and ITIM domain (TIGIT) that interact with their ligands PD-L1, CD80/CD86, MHC-II, and CD155, respectively, on tumor cells [102,103]. This interaction is disrupted with immune checkpoint inhibitors (ICI) and restores the anti-tumor activity of exhausted T cells [102,103]. However, despite the success in malignancies such as melanoma [158], ICI causes adverse side effects [159,160], and targeting the PD-1/PD-L1 axis has not offered any clinical benefit in MM [106], showing high toxicity that led to the halting of different clinical trials [25].

Of interest, anti-PD1 therapy acts on a specific subpopulation of exhausted CD8 TILs termed “progenitor exhausted” cells that retain poly-functionality, persist long term, and differentiate into “terminally exhausted” TILs. Progenitor exhausted CD8 TILs respond to anti-PD-1 therapy and control tumor growth [161]. T cells in MM are highly differentiated [162], which might explain the lack of success in ICI studies. Indeed, within the PD-1 positive population of MILs in MM, only the subset of less differentiated CM T cells are responsive to anti-PD-1, and terminally exhausted T cells associate with worse clinical outcomes after PD-1 inhibition [163]. In addition, the lack of expression of PD-1 in CD8 T cells in newly diagnosed MM patients could also explain the lack of efficacy of the disruption of PD-1/PDL-1 interaction, being TIGIT the most frequent immune checkpoint receptor expressed on T cells [100]. Indeed TIGIT targeting has shown benefits in pre-clinical models of MM [100].

### 3.2. CD4 T Cell Subsets

During MM progression, an alteration in the number and proportion of the different T cell subsets occurs. Specifically, a reduced CD4/CD8 ratio, with lower number of CD4 T cells and Th2 cells is observed [107]. Moreover, IL6 secretion by MM cells [164] and by activated CD3 T cells in MM inhibits the polarization of naïve CD4 T cells into Th1 cells enabling tumor escape [108]. Immunotherapy strategies to obtain a product with an optimized CD4/CD8 T cell ratio [110] could improve this imbalance observed in MM. Indeed a higher CD4/CD8 ratio in the leukapheresis products used to generate CAR T cells in MM correlates with higher responses [111].

Within the CD4 population, regulatory T cells (T-reg cells) are characterized by FOXP3 expression [165]. T-regs have an important immunosuppressive activity towards antigen-presenting cells (APCs) and effector T cells by direct cell–cell contact or soluble mediators that promote immunological tolerance [166]. A variety of studies show that T-regs are enhanced in different types of cancers and related to tumor progression, including hematological malignancies [167]. However, there are conflicting results concerning their number and function in MM [168]. Thus, some studies describe elevated frequencies of functional T-reg cells in newly diagnosed and relapsed patients compared to healthy volunteers [169], while others reported a reduction in their number [170,171], being these T-reg cells dysfunctional [170]. Although there are few studies of T-regs comparing the BM to PB in MM, published reports show that the frequencies of T-regs in both departments are similar and that increased numbers of T-regs in the BM correlate with shorter time to progression and reduced survival [112,113].

The suppressive activity of T-regs in MM has been demonstrated in different studies via IL10 and TGFβ secretion that inhibit dendritic cells and block CD4 T cell-mediated generation of CD8 T cell cytotoxic activity [115]. Inducible T cell co-stimulator (ICOS) and CTLA-4 and expressed by T-regs are involved in their suppressive function [116]. Indeed, MM cells can directly generate functional T-reg cells contact-dependent through ICOS/ICOS-L. These T-regs can be inhibited with anti-ICOS-L MoAb [114]. In MM patients receiving MILs, the baseline percentage of T-regs was lower in patients who achieved a CR compared to those with disease progression. However, on day 360, the percentage of T-regs was normalized, reaching similar levels in all groups, with the CR patients showing the greatest increase [101].

In MM, treatment with talquetamab, a bispecific mAb against the G protein-coupled receptor (GPRC5D), expressed on MM cells, enhanced the anti-MM activity of CD4 conventional T cells but also of T-reg cells. However, T-reg cells presented lower anti-MM activity than conventional CD4 T cells, and seemed to ameliorate the activity of CD4 T cells with decreased production of IFNγ, TNFα, and IL2 [109].

On the other side, some reports demonstrate that higher numbers of T-regs in tumors associate with better prognosis in patients [172], suggesting that the exact prognostic significance of T-regs cells is still unclear, and further studies need to be done. For instance, preclinical studies based on a transient T-reg depletion have demonstrated immune control of MM and prevented disease progression [117].

Th17 cells are pro-inflammatory CD4 T cells that share a common precursor with T-reg cells, where different cytokines will promote the differentiation into one cell subtype or another [173]. Thus, TGFβ favors the formation of T-reg cells. However, in the presence of IL6, IL21 expression is enhanced, activating downstream signaling pathways that, with TGF-β, lead to the differentiation of Th17 cells [118]. In addition, Th17 and T-reg cells can polarize each other [173]. Th17 cells promote MM growth [119] and cause osteoclast-dependent bone damage [174]. Indeed, Th17 cell levels increase further in refractory disease [175] and MBD [176]. Moreover, IL17 production, due to an increased Th17 cell number, induces osteoblast cell death through pyroptosis with the release of IL1β [120] and cooperates with RANKL to further activate osteoclasts enhancing tumor growth and promoting MBD in MM [119]. Newly diagnosed MM patients present a higher number of Th17 cells and IL17 in serum, with decreased T-reg cells. Thalidomide treatment normalized the ratio of Th17 and T-reg cells in PB [121]. Indeed, the efficacy of MoAbs targeting IL17 has been assessed successfully at pre-clinical stages to treat MM [122].

### 3.3. The Impact of Age in T Lymphocytes in MM

The lack of efficacy of ICI in MM has suggested that other dysfunctionalities in T cells might be more relevant in MM, such as the presence of T cell “immunosenescence”. MM is a disease related to the elderly, and the immune system is highly impacted by age. Specifically, “immunosenescence” refers to the deterioration of the immune system that affects T cells, macrophages [177], and natural killer (NK) cells [178], leading to dysfunctional immune responses. In T cells, chronic viral infections, inflammatory diseases, a replacement of BM cellular components by adipocytes, and a thymic involution occurring with aging cause immunosenescence. Immunosenescence is characterized by a loss of the naïve and stem cell memory (SCM) T cell compartments as they differentiate, producing an accumulation of oligo-clonal memory and pro-inflammatory effector T cells that present shorter life and lower proliferative capacity [179,180,181,182]. Immunosenescence also reduces the capacity of T-reg cells to suppress self-reactive T cells and preserve immune homeostasis [180,181]. Moreover, T cell immunosenescence is accelerated with chemotherapy treatments [124] and after therapy with ICIs [125]. In general, most studies agree that immunosenescent T cells lose the naïve CD8 and CD4 T cell compartment, with loss of CD27 and CD28, increased expression of CD57, KLRG1, and PD-1, and secretion of large amounts of IFNγ and TNFα, IL2, IL4, IL10, and IL6 [11,183]. In MM, a higher number of immunosenescent T cells than healthy donors have been described [162] with low proliferative capacity and expression of CD57, KLRG1, CD160, CD28^−^, PD1^low,^ and CTLA4^low^ [123]. Indeed, adoptive cell therapy with MILs in MM showed that the presence of less differentiated CM CD8 T cells correlated with achieving complete responses [101]. Moreover, the pro-inflammatory phenotype of immunosenescent T cells might explain the high number of adverse events observed in MM patients treated with ICIs [25].

In addition, immunotherapy based on administering CAR-T cells has achieved outstanding responses in pediatric patients, where CAR-T cells persist over the years in PB as guardians [22]. However, in MM, patients relapse despite achieving complete responses with a lack of persistence of CAR-T cells [27]. The presence of T cell immunosenescence in MM might partly explain the lack of persistence of CAR-T cells and suggest novel strategies to reverse T cell immunosenescence. The reversal of T cell immunosenescence is a field in development. Different approaches are being tested, including the recombinant growth hormone, that in healthy donors reversed immunosenescence partially with the regeneration of the thymus, a decline in the PD-1 positive CD8 T cells, and an increase in the naïve CD4 and CD8 T cells [184].

Cytokines also influence the differentiation of T cells. In this regard, the production of CAR-T cells and MILs provide a window to add different cytokines and compounds that modify the immunosenescence state of T cells that patients receive. Thus, IL-7, through binding to its receptor IL-7Rα, promotes the differentiation of naïve T cells into effector T cells [185]. Indeed, the addition of IL7 to IL15 during the production of CAR-T cells for MM accelerated the differentiation of CAR-T cells leading to a shorter CAR-T cell persistence in vivo in preclinical models. However, the production of CAR-T cells with IL15 obtained a product less differentiated that demonstrated the highest in vivo persistence compared to IL2 and the combination of IL7 and IL15 [126].

Kinases also regulate T cell senescence, and their inhibition has a high potential to avoid immunosenescence. Thus, inhibition of p38-MAPK in CD4 T cells enhances their survival after TCR activation [186]. Moreover, sestrins bind to and coordinate Erk, Jnk, and p38 MAPK activation in T cells within a complex termed sMAC. Compared to the inhibition of individual MAPKs, sestrin ablation in T cells from old humans disrupted the sMAC complex restoring immune function and antigen-specific functional T cell responses [127]. mTOR pathway is also involved in differentiating naïve CD4^+^ T cells into Th1 or Th17 cells T cells [187]. Thus, in older individuals, mTOR inhibitors, such as rapamycin or everolimus, have improved their immune function [188]. However, rapamycin also suppresses the anti-inflammatory effects of glucocorticoids in human monocytes and myeloid dendritic cells [189]. Inhibition of PI3K during the production of CAR-T cells for MM also provides a less differentiated product associated with a higher duration of response [105]. Adding these compounds only during the production of T cells that patients will receive will avoid the side effects observed after in vivo administration of these drugs.

### 3.4. NK Cells

NK cells are anti-tumor and anti-microbial cells of the innate immune system equipped with a wide array of activating and inhibitory receptors that activate or inhibit their activity through different mechanisms such as granzyme and perforin release, death receptor pathways, and release of additional pro-inflammatory molecules [137,190,191,192,193]. There are two main populations of NK cells in PB, the mature and cytotoxic NK CD56^dim^ and the immature and immunoregulatory CD56^bright^ NK cells [137,192]. Their anti-tumor activity and potential as an allogeneic source of immune cells have made NK cells an attractive target for immunotherapy in different malignancies, including MM [137,194]. A recent meta-analysis study in solid tumors showed that NK cell infiltration associates with increased overall survival in cancer patients. In detail, NK infiltration is more common in earlier stage and higher-grade tumors. Moreover, infiltrating NK cells in intraepithelial regions impacts survival more than infiltrating NK cells in the adjacent stroma [195].

NK cells recognize MM cells through activating receptors, including NKG2D, NKP30, 2B4, NKp80, or DNAM-1, which interact with their ligands in MM cells [137,190]. In newly diagnosed MM patients, the frequency of NK cells in PB does not differ from healthy donors [196]. However, NK cells in MM impair their cytotoxic activity with decreased expression of the activating receptor CD161 and increased expression of the inhibitory CD158a receptor [196]. In addition, MM cells downregulate or block NKG2D and NKp80 on NK cells, inhibiting their activity [128]. NK cells also express PD-1, that in MM patients is up-regulated and interacts with PD-L1 on MM cells [129]. BM stromal cells-derived IL6 inhibits NK cell activity [131] and induces PD-L1 expression in MM cells, impacting the anti-MM activity of NK and T cells [132,133]. Moreover, MDSCs, through tumor-derived IL1β, impair NK cell development and activity [134]. In addition, the expression of chemotactic factors and their receptors in the BM of MM affect the attraction of immune cells. Specifically, NK cells express CXCR3 to enable migration to the BM when required. During MM progression, chemokine ligands involved in the migration of NK cells are imbalanced, including increased levels of CXCL9 and CXCL10 and decreased levels of CXCL12. Moreover, a down-regulation of CXCR3 on NK cells occurs. Altogether drives NK cells outside the BM, weakening their anti-MM activity in the BM [135]. Therefore, altered interactions of NK cells with MM cells impair NK cell anti-tumor activity and are targets for developing strategies that reestablish the anti-MM response of NK cells. Many approaches have been proposed to improve NK cell activity [136,137]. These strategies could be the combination of NK cells with IMiDs and MoAbs that activate antibody-dependent cell cytotoxicity (ADCC) of NK cells. For instance, Daratumumab (anti-CD38), VIS832 (anti-CD138), and Dacetuzumab (anti-CD40) have demonstrated potential in preclinical studies. Bispecific and trispecific killer engagers (BiKEs/TRiKEs) also can redirect NK cells to tumor cells. Antibody recruiting molecules (ARMs) bind a tumor-associated antigen with endogenous IgG that will induce NK-mediated ADCC. NK cell activators, such as ALT-803, an IL-15 superagonist, stimulate NK cells and T cells, and finally, CAR-NKs targeting SLAMF7, CD138, or NKG2D ligands MM antigens have demonstrated pre-clinical efficacy [136,137].

In addition, decidual NK (dNK) cells represent a transient population present during the first months of pregnancy. dNK cells are CD56^bright^ and highly angiogenic through the production of proangiogenic factors, including VEGF, PlGF, CXCL8, IL10, and angiogenin [197,198,199]. It has been suggested that the recruitment of conventional CD56^bright^ PB-NK cells could contribute to their origin [200,201]. This angiogenic or “nurturing” role of dNK cells required during early pregnancy presents homologies to the angiogenic processes during tumor progression. In this regard, in relapsed/refractory and newly diagnosed MM patients, the CD56^bright^ NK cell subset, highly activated, is the prevailing NK cell subset in both BM and PB, being these differences more pronounced in the BM [130]. Therefore, further studies in MM patients should add additional markers to characterize better a possible angiogenic activity of CD56^bright^ NK cells detected in the BM of MM patients. Of interest, in vitro expanded NK cells administered in immunotherapy acquire a CD56^bright^ phenotype [202,203]. Thus, a possibility to improve clinical studies that infuse expanded NK cells in MM patients could be the previous selection of CD56^dim^ mature NK cells.

### 3.5. Regulatory B Cells

B-reg cells represent another essential immunosuppressive arm in the adaptive immune system that is affected by their interaction with MM cells. B-reg cells maintain immune tolerance and suppress autoimmune and inflammatory responses through secretion of IL35, TGFβ, and IL10, as well as expression of inhibitory molecules, such as PD-L1 [204]. In healthy donors, IL10 secretion of B-reg cells suppresses CD4 T cell proliferation and the release of IFN-γ and TNF-α by CD4 T cells, inhibits CD4 T cell differentiation into Th1 and Th17 cells, and favors CD4 T cell polarization into T-regs [138]. In the TME, tumor-released autophagosomes induce this B-reg population [205]. Despite their poorly described role in MM, B-reg cells have become essential players increasing during the initial stages of MM and head for extinction in relapsed or refractory MM [206]. Thus, newly diagnosed MM patients have no differences in PB B-reg cells compared to healthy donors. However, B-reg cells accumulate in the BM of newly diagnosed MM patients rather than in PB, where MM cells promote the survival of B-reg cells. Moreover, B-reg cells in PB and BM are higher at MM diagnosis than after response to therapy, and, finally, at relapse, they are too low to be detected. Moreover, MM B-regs avoid NK-ADCC against MM cells [139]. Despite the relevant role of B-reg cells in the progression of MM, strategies to target them in the clinic have not been described yet. Novel research to decipher cellular interactions of B-reg cells with other cells and how B-reg cells exert their suppressive activity is required first to foresee the possible implications of their targeting.

### 3.6. Tumor-Associated-Macrophages (TAMs)

Tumor-associated-macrophages (TAMs) abundant in the MM microenvironment, can protect MM cells from chemotherapy-induced apoptosis and exert anti-tumor activity [207,208]. Even though macrophages present high plasticity, they are traditionally classified into: (1) M1 activated macrophages with anti-tumor activity that secrete pro-inflammatory cytokines; or (2) the alternatively activated M2 macrophages with immunosuppressive activity. Indeed, several clinical studies in MM patients have associated a high infiltration of M2 macrophages in the BM with poor prognosis and poor response to current treatments, while patients with increased infiltration of M1 macrophages showed better outcomes [209]. M1/M2 polarization will rely on signals present in the environment. Thus, pro-inflammatory cytokines promote M1 differentiation, and immunosuppressive ones drive the differentiation towards an M2 phenotype [210]. MM cells influence the phenotype and recruitment of macrophages in the BM. Thus, CXCL12 production by MM cells and BM-MSCs increases monocyte recruitment through the CXCL12/CXCR4 axis and induces M2 macrophage polarization. These CXCR4-M2-educated macrophages are grown in the BM of MM compared to healthy controls, support MM cell proliferation, protect them from chemotherapy, and suppress T-cell proliferation and activity [140]. Of interest, MM cells secrete extracellular vesicles (EVs) that polarize recruited monocytes to an M2 phenotype with subsequent release of M2-associated cytokines such as IL6, IL10, IL8, and TNFα that promote tumor proliferation. Circulating miR-16 in serum impairs these events. Indeed, high miR-16 levels associate with prolonged survival in MM patients [211]. Moreover, soluble secreted forms of M2 macrophage receptors, such as CD206 and CD163, and chemokines like CCL2, have been proposed as biomarkers for disease progression, prognosis, and treatment response [141,142]. Moreover, macrophages directly affect immune evasion using their macrophage immune checkpoint. The binding of CD47 in the membrane of MM cells to signal-regulatory protein alpha (SIRPα) receptor on the surface of macrophages leads to downstream signaling within the macrophages, resulting in inhibition of phagocytosis activity [143].

The relevance of macrophages in MM is evidenced by increasing strategies used in preclinical and clinical studies to counteract their supportive role in MM progression. As mentioned, the CXCR4/CXCL12 axis contributes to MM cell adhesion and migration and promotes monocyte recruitment and differentiation towards a proangiogenic and immunosuppressive M2-like phenotype. In preclinical studies, the inhibition of CXCR4 significantly suppressed monocyte recruitment to the BM [140]. For that reason, analyzing the impact of AMD3100 (CXCR4 inhibitor) on macrophages in MM patients could reveal positive results. Once in the BM, depletion of BM resident macrophages is a strategy to reduce macrophage tumor support with promising preclinical results, proving that macrophage depletion can limit MM disease burden [144]. Other targeted therapies block the pro-tumor functions of TAMs, reprogramming them to reduce their immunosuppressive M2 phenotype and promote M1 phenotype. Currently, the most common approach for targeting macrophages involves inhibiting the CSF1 receptor (CSF1R), which regulates the migration, differentiation, and survival of macrophages and their precursors [145]. In MM, a recent preclinical study has demonstrated a significant reduction in tumor burden following treatment with the anti-CSF1R antibody [212], suggesting that targeting macrophages, via CSF1R, in combination with standard therapies, may be a promising therapeutic strategy in MM. Finally, targeting immune checkpoints in macrophages has been an emerging topic [213]. Currently, some clinical studies aim to inhibit the CD47-SIRPα immune checkpoint using various strategies, including anti-CD47 antibodies (SRF231: NCT03512340 and AO-176: NCT04445701) or SIRPα-IgG1 Fc fusion proteins (TTI-621: NCT02663518 and TTI-622: NCT03530683). In general, TAM-targeting therapy represents a promising treatment for MM patients and could improve current MM cell therapies, overcoming unresponsiveness and drug resistance.

### 3.7. Myeloid-Derived Suppressor Cells

(MDSCs) are a heterogeneous subset of immature myeloid progenitor cells that inhibit innate and adaptive immune responses, induce T-reg differentiation, promote angiogenesis, and even differentiate themselves into functional osteoclasts, promoting tumor growth [28]. MDSCs can be further divided into CD14+ monocytic (M-MDSC) and CD15+ granulocytic (G-MDSC) subsets being G-MDSC increased in the PB of MM patients [214]. Murine models show that MDSCs accumulate in the BM during MM progression in the early stages of the disease while circulating myeloid cells increase at later stages, and MDSC targeting reduces tumor load [215]. Moreover, the frequency of M-MDSCs in PB predicts outcomes after lenalidomide-dexamethasone treatment, where failure to achieve a response associates with an increase in M-MDSC frequency after treatment [216].

MDSCs perform an important suppressive activity of CD8 T cells and NKT cells in the BM. To achieve these suppressive functions, MDSCs produce arginases (ARG1), reactive oxygen species (ROS), cyclooxygenase-2 (COX2), inducible NOS (iNOS), IL6, IL10, IL18, and reduce metabolic factors required for T-cell activation. Moreover, MM cells induce MDSCs development from PB mononuclear cells in healthy donors, suggesting the relevance of bi-directional cell–cell communication [146]. In addition, soluble factors from MM cells, such as IL10, CCL5, MIP-1 or large amounts of IL6 generate MDSCs with T cell suppressive ability through Mcl-1 upregulation that enhances MDSCs survival [215]. In addition, BM stromal cells-derived exosomes in MM induced survival of MDSCs through STAT3 and STAT1 pathways and increased anti-apoptotic proteins Bcl-xL and Mcl-1 [217].

The relevance of MDSCs has been translated to patients. Thus, dual targeting of MM cells and MDSCs with biotherapeutic agents, such as Daratumumab [147], has emerged as a promising new therapeutic option with high potential, as recently reviewed by Uckun [218]. On the other hand, the α-chain of the IL3 receptor, known as CD123, is highly expressed on MDSCs [219]. For that reason, several biotherapeutic agents targeting CD123 have been developed, including the CD123-directed recombinant human IL3 fusion toxin Tagraxofusb (SL-401). SL-401 has been assessed in combination with the standard of care (NCT02661022) in relapsed/refractory MM patients with promising early evidence of clinical activity. Altogether, these studies suggest that therapies targeting MDSCs might overcome the immunosuppressive environment in the BM of MM and increase the anti-tumor effect of additional treatments for MM patients.

## 4. Conclusions

To conclude, tumor cells in MM develop a smart network in the BM with non-hematological and hematological cells that create an environment that nurtures and protects them from traditional chemotherapy and novel based-immunotherapy treatments. Interactions with cells that do not belong to the hematological BM compartment form a feedback loop that leads to bone destruction, angiogenesis, and tumor expansion in the BM. Moreover, whereas immune cells are supposed to perform tumor surveillance, MM cells and non-hematological cells in the BM can polarize the phenotype and activity of immune cells leading to immune evasion. Unraveling these interactions has allowed the development of novel treatments for MM patients. In addition, MM patients represent an elderly population with immunosenescent T cells, which might explain the lack of efficacy of some immunotherapy treatments.

Finally, finding the most appropriate treatment for each MM patient among all the available treatments is an urgent need that requires first to have a global picture of the proportion of the different cell subsets in the TME. Deciphering the cellular imbalance in the TME will help select the best treatment combination that could restore this imbalance for each patient. Indeed, novel technologies based on multi-immunofluorescence allow deciphering cell–cell interactions in the TME and the phenotype of cells that will help treat patients.

## Figures and Tables

**Figure 1 cancers-14-03796-f001:**
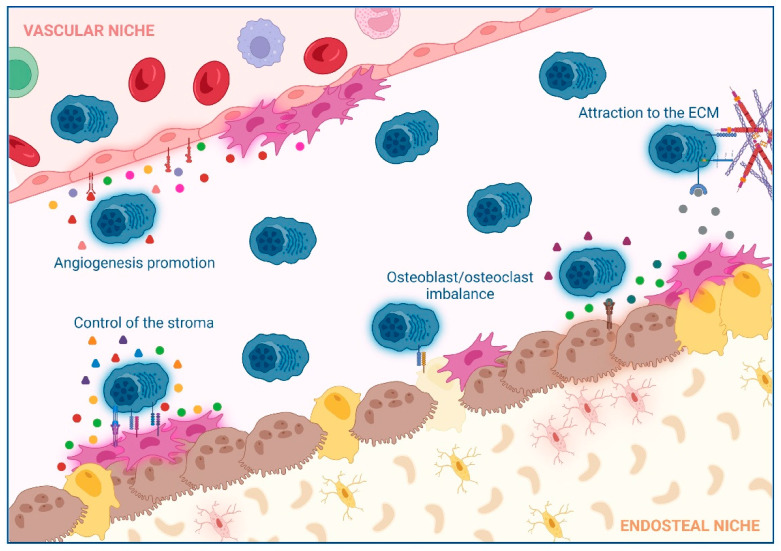
**Summary of the interactions between non-hematological cells and multiple myeloma (MM) cells in the bone marrow (BM)**: Different cell populations interacting with MM, receptors involved and secreted molecules by the different cell subsets that impact MM cell proliferation are indicated. The extracellular matrix (ECM) causes an attraction of MM cells to the BM. Bone marrow mesenchymal stromal cells (BM-MSCs) and MM cells interact, making the stroma a favorable environment for MM cells. MM cells and BM-MSCs alter the balance between osteoblast formation and osteoclast degradation. Endothelial cells enhance the angiogenesis in the BM to favor extramedullary disease.

**Figure 2 cancers-14-03796-f002:**
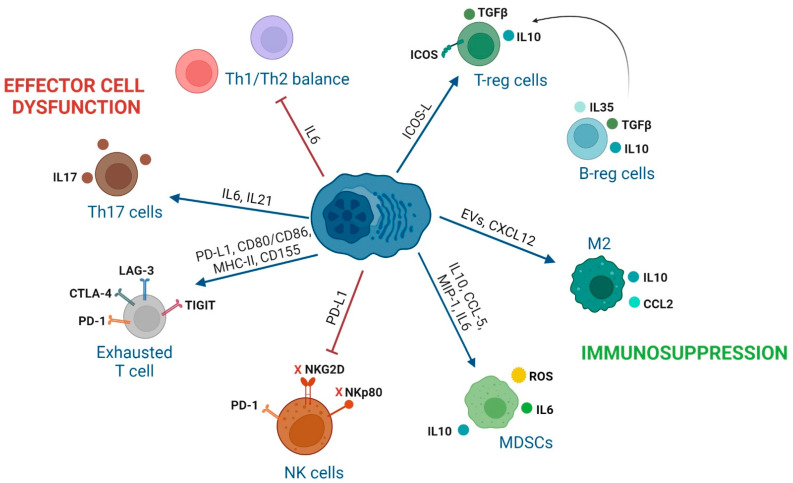
**Summary of the impact that secreted molecules or expression of receptors by MM cells cause on the polarization and activity of immune cells.** MM cells directly generate functional T regulatory (T-reg) cells contact-dependent by ICOS/ICOS-L. IL6 and IL21 secretion in the BM causes a decrease in the T helper (Th)1 cell populations leading to an imbalanced Th1/Th2 ratio. IL6 and IL21 secretion by MM cells enhance the production of Th17 cells. Over-expression of ligands of immune-checkpoint receptors in T cells causes exhaustion of T cells and natural killer (NK) cells. Secretion of IL10, CCL5, MIP-1, and IL6 from MM cells generates MDSCs with T cell suppressive ability. CXCL12 production and secretion of extracellular vesicles (EVs) by MM cells increases monocyte and induces M2 macrophage polarization.

**Table 1 cancers-14-03796-t001:** MM the cell–cell interactions and molecules involved in the interactions that may affect disease progression and the anti-MM therapies that could target these interactions.

Cellular Compartment or Process	Molecules and/or Cell Population Involved	Impact on MM Disease	Therapeutic Strategy Proposed
ECM	1. CXCR4/CXCL12. 2. CD138 and VLA-4 (MM)/Fibronectin (ECM).	1. MM homing into the BM [30]. 2. NFkB activation, tumor survival, drug resistance [32].	1–2: AMD3100 (CXCR4 inhibitor), and Bortezomib (VLA-4 downregulation) [34].
BM-MSCs	1. VLA-4 (MM)/VCAM-1 (BM-MSCs). 2. LFA-1 (MM)/ICAM-1 (BM-MSCs). 3. IL6 secretion by BM-MSCs induced by MM cells. 4. Notch pathways and DKK1. 5. LINC00461 in BM-MSCs exosomes 6. Ligation of BAFF. 7. APRIL secretion (BM-MSCs)/BCMA and TACI (MM) [51] 8. TNFα.	1. NFkB activation, MM survival [39]. 2. Disease progression [40]. 3. Enhanced secretion of VEGF and bFGF by MM that re-stimulates IL6 production [43]. 4. IL6, VEGF, and IGF-1 secretion in BM-MSCs [46,47]. 5. MM cell proliferation and drug resistance [98]. 6–7. MM proliferation [50]. 8. Enhanced LFA-1, ICAM-1, VCAM-1, and VLA-4 (MM) and ICAM-1 (BM-MSCs), increased binding of MM to BM-MSCs and further IL6 secretion [54].	1. Natalizumab: anti-α4 integrin (NCT00675428). 2. LFA878: LFA-1 inhibitor (preclinical studies) [41]. 3. Tocilizumab: anti-IL6R [57]. 4. BHQ880: anti-DKK1 [58]. 5. LINC00461 knockdown (preclinical studies) [98]. 6. Tabalumab: anti-BAFF [59]. 7. APRIL-based CARs target BCMA or TACI [52]. 8. Anti- TNFα. However, these drugs in other inflammatory conditions increase the risk of MM [56].
Osteoclast/osteoblast imbalance	1. MIP1α and MIP1β (MM). 2. RANKL (osteocytes)/RANK (osteoclasts). 3. MM induce RANKL and IL6 secretion by BM-MSCs. 4. VLA-4 (MM)/VCAM-1 (osteoblasts and BM-MSCs).	1. Osteoclast activation [63,64], IL6 secretion [65], CHSY1 up-regulation, Notch signaling, MM survival, recruitment of osteoclast precursors [66]. 2–3. Osteoclast activity [67,68]. 4. RUNX2 decreased activity, decreased osteoblast differentiation [72], decreased OPG secretion, osteoclast formation and bone degradation [73].	1-2-3. Amino-bisphosphonates that inhibit osteoclast activity [69]. 2–3. Denosumab: anti-RANKL [70] 4. Natalizumab: anti-α4 integrin (NCT00675428). 4. BHQ880: anti-DKK1 [75].
Angiogenesis in the vascular niche	1. VEGF production (MM). 2. EGFR-2, Tie2/Tek, β3-integrin and endoglin in MM endothelial cells. 3. MM cells induce HGF, VEGF and IL8 secretion in BM-MSCs. 4. IGF1 and IL6 secretion by MM endothelial cells.	1. Angiogenesis [82]. 2. Enhanced MM cell interaction with new blood vessels and further dissemination [83]. 3. Neovascularization [89]. 4. MM growth, enhanced MM production of VEGF, PDGF, Ang-1, HGF, and IL1. Enhanced angiogenesis [90].	1. Amino-bisphosphonates are anti-angiogenic [69,99]. 1–3. Bevacizumab: anti-VEGF [92,93]. 2. Derivatives of quinolone and quinazoline inhibit VEGFRs, EGFR, and PDGFR [94,95]. 4. Immunomodulators [96,97].
Effector CD8 T cells	1. CXCR4 (MILs)/CXCL12 (BM-MSCs). 2. CM phenotype of MILs. 3. PD-1, CTLA-4, LAG-3, or TIGIT (T cells) with PD-L1, CD80/CD86, MHC-II, and CD155 (MM). 4. TIGIT expression on T cells in MM [100].	1. Trafficking of MILs to the BM [101]. 2. Enhanced CR in patients [101]. 3. Inhibition of T cell activity [102,103] 4. Dysfunctional T cells with decreased proliferation and cytokine production [100].	1. Administer MILs with enhanced CXCR4 expression that has shown efficacy in CAR-T cells [104]. 2. Addition of PI3K inhibitors during the production of MILs [105]. 3. ICI treatments targeting others than PD-1/PD-L1 due to their toxicity in MM [106]. 4. TIGIT inhibition [100].
CD4 conventional T cells	1. Reduced CD4/CD8 ratio, lower number of CD4 T and Th2 cells in MM [107]. 2. IL6 secretion inhibits polarization of naïve T cells into Th1 cells [108]. 3. GPRC5D (MM)/CD4 T cells [109].	1–2. Tumor escape to immune surveillance [108]. 3. Inhibition of CD4 T-cell anti-MM activity.	1. Optimization of CD4/CD8 ratio in cellular immunotherapy products [110,111]. 2. Tocilizumab (anti-IL6R). 3. Bispecific antibody against GPRC5D. (talquetamab) enhances anti-MM activity of CD4 T cells [109].
T-reg cells	1. Increased T-regs in the BM of MM [112,113]. 2. IL10 and TGFβ secretion by T-regs. 3. CTLA-4 and ICOS expression in T-regs. 4. ICOS (T-reg)/ICOSL (MM) [114]. 5. GPRC5D (MM)/T-regs [109].	1. Shorter time to progression [112,113]. 2. Interruption of CD4 T cell-mediated generation of CD8 T cell responses [115] 3. T-reg suppressive activity [116]. 4. Generation of functional T-regs [114]. 5. Inhibition of CD4 conventional and T-reg activity.	1–2. Optimized MIL product with lower number of T-regs induces CR [101]. 2. Transient T-reg depletion [117]. 3, 4. Inhibition of T-regs with anti-ICOSL MoAb [114]. 5. Talquetamab enhances anti-MM activity of T-reg cells by themselves [109].
Th17 cells	1. IL6 induces IL21 that with TGFβ induces Th17 differentiation [118].	1. MM growth [119], osteoblast cell death [120], osteoclasts activation, tumor growth and MBD [119].	Thalidomide normalizes the ratio of Th17 and T-reg cells in PB [121]. Anti-IL17 Ab show anti-MM activity [122].
Age in T cells	High number of immunosenescent T cells (CD57, KLRG1, CD160, CD28^−^, PD1^low^, and CTLA4^low^) [123].	Enhanced by chemotherapy [124] and ICI treatments [125].	Addition of PI3K inhibitors [105], IL15 [126] and sestrins inhibition [127] during the production of the immunotherapy product.
NK cells	1. MM cells downregulate NKG2D and NKp80 on NK cells [128]. 2. PDL1 (MM)/PD1 (NK cells) [129]. 3. BM-MSCs-derived IL6. 4. Tumor-derived IL1β in MDSCs. 5. Increased CXCL9 and CXCL10, decreased CXCL12, down-regulation of CXCR3 on NK cells. 6. CD56^bright^ NK cells highly activated in BM and PB [130].	1. Inhibition of NK activity [128]. 2. Inhibition of NK activity [129]. 3. NK inhibition [131], PD-L1 on MM cells, impacting the NK and T cell activity [132,133]. 4. NK inhibition [134]. 5. Driving of NK cells outside the BM [135]. 6. Additional markers to characterize a possible angiogenic activity of CD56^bright^ NK cells.	1-2-3-4-5: Combination of IMiDs and MoAb enhance endogenous NK cell activity and ADCC of NK cells. BiKEs/TRiKEs redirect endogenous NK cells to tumor cells. Ab recruiting molecules bind tumor-associated antigens with endogenous IgG inducing NK-mediated ADCC. ALT-803: IL-15 superagonist that stimulates NK cells and T cells. CAR-NKs targeting SLAMF7, CD138 or NKG2D ligands on MM [136,137]. 6. Previous selection of in vitro expanded CD56dim NK cells.
Regulatory B cells	MM cells promote B-reg cell survival and their accumulation in the BM.	1. IL10 secretion of B-reg cells inhibits CD4 T cell differentiation into Th1 and Th17 cells, and favors polarization into T-regs [138]. 2. B-regs avoid NK-ADCC in MM [139].	Strategies to target B-reg cells have not been described yet. Novel research to decipher cellular interactions with B-regs and how B-regs exert their suppressive activity is required.
TAMs	1. CXCL12 (MM and BM-MSCs)/CXCR4 (monocytes). 2. M2 macrophage immunosuppresion through IL6, IL10, IL8, TNFα, CD206, CD163, CCL2. 3. CD47 (MM)/SIRPα (macrophages).	1. Monocytes recruitment and M2 polarization in BM [140]. 2. MM proliferation and progression [141,142]. 3. Immune checkpoint resulting in a “don’t eat me” signal in M2 macrophages and immune evasion [143].	1. AMD-3100: CXCR4 inhibitor (preclinical studies) [140]. 2. Clodronate liposome to deplete resident M2 macrophages in BM (preclinical studies) [144]. 2. Anti-CSF1R to reprogram TAMs to promote M1 phenotype (preclinical studies) [145]. 3. Antibodies anti-CD47 (SRF231: NCT03512340 and AO-176: NCT04445701). 3. SIRPα-IgG1 Fc fusion proteins (TTI-621: NCT02663518 and TTI-622: NCT03530683).
MDSCs	1. IL10, CCL5, MIP-1 or IL6 from MM cells generate MDSCs 2. ARG1, ROS, COX2, iNOS, IL6, IL10 and IL18 (MDSCs)	1–2. Inhibit immune responses, induce T-regs, promote angiogenesis and differentiate into osteoclasts [146].	1. Daratumumab: anti-CD38 (dual targeting of MM cells and MDSCs) [147] 2. Tagraxofusb: CD123-targeted agent [148]

ECM: extracellular matrix; BM-MSCs: Bone marrow mesenchymal stromal cells. TAMs: tumor-associated macrophages. MDSCs: myeloid-derived suppressor cells. MM: multiple myeloma. BM: bone marrow. PB: peripheral blood. MILs: marrow-infiltrating lymphocytes. ICI: immune checkpoint inhibition. MBD: myeloma bone disease. MoAb: monoclonal antibody. CM: central memory. CR: complete response. ADCC: antibody-dependent cell cytotoxicity. Ang-1: Angiopoietin 1; APRIL: A Proliferation-Inducing Ligand; ARG-1: Arginase 1; BAFF: B-Cell Activating Factor; BCMA: B Cell Maturation Antigen; bFGF: basic Fibroblast Growth Factor; CCL2: C-C motif chemokine Ligand 2; CCL5: C-C motif chemokine Ligand 5; CHSY1: Chondroitin Sulfate Synthase 1; COX2: cyclooxygenase 2; CTLA-4: Cytotoxic T-Lymphocyte-associated Antigen 4; CXCL9: C-X-C motif chemokine Ligand 9; CXCL 10: C-X-C motif chemokine Ligand 10; CXCL12: C-X-C motif chemokine Ligand 12; CXCR3: C-X-C motif chemokine Receptor 3; CXCR4: C-X-C motif chemokine Receptor 4; DKK1: Dickkopf1; EGFR-2: Epidermal Growth Factor Receptor 2; GPRC5D: G Protein–coupled Receptor, class C, group 5, member D; HGF: Hepatocyte Growth Factor; ICAM-1: Intercellular Adhesion Molecule 1; ICOS: Inducible T-cell COStimulator; IGF-1: Insulin-like Growth Factor 1; IL: interleukin; iNOS: inducible Nitric Oxide Synthase; KLRG1: Killer cell Lectin-like Receptor subfamily G member 1; LAG-3: Lymphocyte Activation Gene 3; LFA-1: Lymphocyte Function-associated Antigen 1; MHC-II: Major Histocompatibility Complex class II; MIP1α: Macrophage Inflammatory Protein 1 α; MIP1β: Macrophage Inflammatory Protein 1 β; NFkB: Nuclear Factor kappa-light-chain-enhancer of activated B cells; NKG2D: Natural Killer Group 2 member D; OPG: Osteoprotegerin; PD-1: Programmed Death 1; PD-L1: Programmed Death-Ligand 1; RANK: Receptor Activator of Nuclear Factor k B; RANKL: Receptor Activator of Nuclear Factor k B Ligand; ROS: Reactive Oxygen Species; RUNX2: Runt-related transcription factor 2; SIRPα: Signal Regulatory Protein α; TACI: Transmembrane Activator and CAML (Calcium-Modulator and Cyclophilin Ligand) Interactor; TGFβ: Transforming Growth Factor β; TIGIT: T cell Immunoreceptor with Ig and ITIM domains; TNFα: Tumor necrosis factor α; VCAM-1: Vascular Cell Adhesion Molecule 1; VEGF: Vascular Endothelial Growth Factor; VLA-4: Very Late Antigen-4.

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
