# Peer review of "A Journey through the Inter-Cellular Interactions in the Bone Marrow in Multiple Myeloma: Implications for the Next Generation of Treatments"

_cancers, 2022, doi:10.3390/cancers14153796_

Round 1
Reviewer 1 Report
Thank you very much for giving me the opportunity to review this interesting review manuscript on cell-cell interactions in MM.
This manuscript is a review to describe the interactions of MM cells with cells of the other compartment too deep in their impact on MM progression and identify new therapeutic targets. It is relevant in terms of the compilation of the studies carried out in each of the sections established by the authors in the different cellular compartments, This facilitates the understanding of the pathophysiology in a neoplastic complex disease. There are no publications of this type that collect all the interactions in a single article, this is an ambitious project, the question is whether all relevant publications are compiled in this review? To be strict in the proposed review, a systematic search should be made following the PRISMA 2020 guidelines in order not to leave any relevant work out of the scope with inclusion criteria appropriate to the objective of the review and a listing of the directories reviewed. The conclusions summarize the pathophysiological interpretation by the authors of what occurs in the bone marrow and what produces an advantage in proliferation and diffusion to plasma cells as well as resistance to therapies developed in recent years. It is missing that as a consequence of all the review carried out, the authors do not propose a personalized combination therapeutic strategy based on the individual characteristics of the patients in relation to the predominance of cellular interaction or how to circumvent the cellular interaction. The references are complete and include the doi which in principle is not required by the journal. The manuscript has no tables. The two figures summarize well the interactions between plasma cells and non-hematological cells and between immune cells in bone marrow. They seem to be the authors' own interpretations.
In a general reading of the work, two aspects stand out: 1. the fact that a systematic review has not been carried out following the PRISMA recommendations, perhaps this was not the intention of the authors of the review, and 2. the second part of the title of the manuscript refers to "their implications for designing novel treatments" do not go into more depth in each section on possible combinations of therapeutic strategies. A suggestion could be to dedicate a final section with its own contribution as an example of how to measure these factors involved in the aggressiveness of the disease or in the lack of response to some treatments. in routine clinical practice in order to identify in each case what could be the ideal personalized therapeutic combination in this complex and incurable entity such as MM.
Reviewer 2 Report
The manuscript "A journey through the inter-cellular interactions in the bone marrow in multiple myeloma and their implications for designing novel treatments " is well written review on an interesting topic of new research directions in MM biology and therapy.
The Authors carefully and in depth described the network of interactions between MM cells and the tumor microenvironment.
However, I have the feeling that there are many works on similar topics.
The text is quite heavy. Data on novel therapies designed based on biological roles in the bone marrow are poorly emphasized. I propose to add a summary table with their examples that would help to make the work more attractive for the Readers.
Reviewer 3 Report
Innovative approaches for treating Myeloma patients that inevitably become refractory to prior lines of therapy is a clinically relevant topic. Here, the authors elegantly address the distinct intercellular interactions occurring between myeloma plasmocytes and both non-hematopoietic and hematopoietic components at the bone marrow niche during the progression and development of resistance to therapy. The manuscript is very well written and organized following a biological storytelling that is very pleasing to follow. Although the present form of the manuscript has excellent quality to be published in Cancer journal the reviewer finds that a small discussion section plus a summarizing table (e.g. prior conclusion) on the new anti-myeloma molecules/therapies that target the tumour microenvironment (that are in clinical trials or eventually others that the authors deemed relevant to follow over the next decade) would turn this review manuscript into an outstanding one. The reviewer congrats the authors for the excellent review on this topic.
Minor issues:
Q1: The reviewer suggests to change the title to: “A Journey through the inter-cellular interactions in the bone marrow in multiple myeloma: implications for the next generation of treatments”
Q2: Remove paragraph at lines 89-91, typo?
Q3: Remove underline at line 337.
Round 2
Reviewer 2 Report
Dear Authors,
Thanks for your effort to review the paper, it have brought an improvement of quality of work.